# A Pilot Study on Pharmacists’ Knowledge, Attitudes and Practices towards Medication Dysphagia via Asynchronous Online Focus Group Discussion

**DOI:** 10.3390/ijerph20042858

**Published:** 2023-02-06

**Authors:** Poh Leng Tan, Terence JunHeng Loh, Sui Yung Chan

**Affiliations:** Department of Pharmacy, National University of Singapore, 18 Science Drive 4, Singapore 117559, Singapore

**Keywords:** medication, dysphagia, knowledge, attitude, practices, focus group discussion, medication safety

## Abstract

Medication dysphagia (MD) refers to difficulty swallowing oral medications. To cope, patients may inappropriately modify or skip medications, leading to poorer outcomes. Little is known about healthcare professionals’ (HCPs’) perspectives in managing MD. This study investigated pharmacists’ knowledge, attitudes, and practices (KAP) in caring for patients with MD. An asynchronous online focus group was pilot tested in seven pharmacists, with up to two questions posted daily on an online platform over 15 days. Thematic analysis of the transcripts revealed five interrelated themes: (1) knowledge about MD; (2) management of MD; (3) expectations of patient proactivity; (4) desire for objectivity; (5) professional roles. The findings provided insight into pharmacists’ KAP and may be incorporated into a full-scale study involving various HCPs.

## 1. Introduction

Solid oral dosage forms (SODFs) are common due to their convenience and ease of accurate dosing [1]. Their use is expected to increase alongside growing chronic disease burden in an ageing society [2,3]. Therapeutic efficacy depends on the patient’s ability to swallow SODFs. However, 10–40% of patients reportedly experienced medication dysphagia (MD) [4,5], defined as the subjective sensation of difficulty swallowing oral medications, solid or liquid, even in otherwise healthy persons. The scope of this study is limited to SODFs due to its predominance [1].

There are two distinct terms that overlap with MD—clinical dysphagia (CD) and medication-induced dysphagia (MID). CD is the difficulty or disability transferring solids or liquids from the mouth to the esophagus due to the dysfunction of one or more of the physiological processes involved in swallowing [5,6,7,8]. Persons with MD do not necessarily have CD and vice versa, although they may coexist [4,5,9,10,11,12,13]. This distinction is supported by opposing epidemiological trends [4,14] and different reported causes [1,15]. The prevalence of MD peaks early and declines with age; on the other hand, the prevalence of CD increases with age [4,14]. The most common causes of MD and CD are aversion to medications [1] and neurological conditions [15], respectively. Regardless, they appear inconsistently and are easily misdiagnosed, resulting in unnecessary interventions in patients with CD who are capable of swallowing SODFs [16,17]. MID, on the other hand, is CD resulting from side effects of medications such as olanzapine and quetiapine [8,15].

A review of the literature revealed frequent overlaps in recommendations for MD management, with dosage form modification (DFM) consistently being the last choice [4,5,18,19]. For the purposes of this study, common recommendations have been summarized into an MD management plan (Figure 1) to act as a reference point for comparison with current practices.

DFM includes, but is not limited to, cutting, crushing, or chewing tablets, opening capsules, and mixing SODF contents with fluids for swallowing or administration via enteral tubes; it excludes instances where DFM is by design, such as chewable or effervescent tablets. DFM may alter a medication’s pharmacokinetic profile and efficacy, increase the risk of adverse effects, or result in incomplete dosing [4,5,18,20]. Persons modifying cytotoxic or hormonal medications without adequate protection are also exposed to hazards [5,20]. Since DFM constitutes off-label use, healthcare professionals (HCPs) are liable for any resultant harm that patients may suffer [20,21].

Despite the availability of MD management strategies, its management by HCPs is inadequate [14]. Studies in community settings across various countries showed that MD was largely unaddressed [1,13,14,19] as most patients were not asked about difficulties with swallowing medications by HCPs; some HCPs did nothing even when notified by patients. DFM appeared to be HCPs’ default option to address MD despite suitable alternative drugs or dosage forms being available [4,20]. Studies showed that among HCP-modified medications, up to 32% were unsuitable for modification [19,22] and 44% were without prescribers’ knowledge [19,20].

MD is likely to remain unaddressed without HCPs’ proactivity. Studies consistently found that most patients would not voluntarily inform HCPs about MD [1,4,13,23]. To cope with MD, patients may adopt inappropriate strategies. A study found that 58.8% of patients turned to DFM, despite 49.4% of them being unaware of its potential implications [1], which include the aforementioned risks, and even death [14,16]. Alternatively, up to 68.7% of MD patients resorted to non-adherence [24], affecting clinical outcomes, increasing morbidity, mortality, and unnecessary healthcare costs. Non-adherent patients have twice the mortality risk of their adherent counterparts [19]. Those who remain adherent without DFM face the risk of choking, lodging of SODFs in the esophagus, and mucosal injury [9]. Hence, HCPs’ proper management of MD is crucial.

Studies on MD have focused on patients’ perspectives without expanding on HCPs’ knowledge, attitudes, and practices (KAP). A KAP study can reveal what is known (knowledge), believed (attitude), and done (practiced) from HCPs’ perspectives in the context of MD. It provides information for better understanding on the topic of MD, as well as identifies needs and barriers which help in the development and implementation of interventions to address MD [25]. Pharmacists’ roles in the management of MD include assessing and advising of the suitability of modifying SODFs, as well as proposing alternatives to SODFs that are not suitable for modification. This study aimed to conduct a pilot study on pharmacists to investigate their KAP in caring for patients with MD.

## 2. Materials and Methods

This study adopted an inductive qualitative study design to investigate the KAP of pharmacists in caring for MD patients, where existing knowledge is scant. An asynchronous online focus group (AOFG) was utilized to elicit a breadth of responses through participant interactions. Although typically used when studying vulnerable persons or sensitive topics, its adoption here was deemed advantageous for various reasons [26,27,28] (Table 1).

### 2.1. Ethics

Ethics approval was obtained from the National University of Singapore Pharmacy Ethics Committee (PHA-DERC-14).

### 2.2. AOFG Platform

Platforms from published papers [28,30,31,32,33,34,35] and Google [36,37,38,39,40,41,42,43,44,45,46,47,48,49,50] were identified and evaluated. A set of criteria (Appendix A) was adapted for evaluating platforms [51]. FocusGroupIt was chosen for fulfilling key criteria, acceptable pricing scheme, and ease-of-use.

### 2.3. AOFG Discussion Guide

The development of the discussion guide was shaped by prior literature review of MD [1,4,5,9,10,12,13,14,15,16,17,18,19,20,21,23,24]. Question sequencing was based on flow of ideas and progressive complexity [52]. Upfront provision of guiding and probing questions provided participants with direction and ample opportunity to develop forthright responses [28]. A scenario-based question about patients experiencing MD, with a follow-up question revealing their recent history of stroke, was included to understand how pharmacists would respond when presented with patients with MD and medical history of varying severities. Questions were internally refined, and then reviewed by four experienced pharmacists for face and content validity.

### 2.4. Participants and Recruitment

Pharmacists were recruited via convenience sampling. Inclusion included current involvement in direct patient care (i.e., those working in nursing homes, hospitals, or the community) and possession of a device with internet access. Data saturation was not a consideration in this pilot study.

Prospective participants were contacted via email. Participants’ consent and contact details for dissemination of invitation links and reminders were collected via email replies. Subsequently, participants were emailed invitation links and notified of the start date. Of the 9 pharmacists who responded, 2 declined, owing to their busy work schedule. A prior relationship had not been established with the moderator, LJT.

### 2.5. Procedure

The study protocol was developed based on common practices and guiding principles outlined in the existing literature [26,27,28]. Seven participants were recruited. This number was within the ideal range (6–8) of a group size [26,28] to optimize participant interaction and produce manageable volumes of data [28]. Homogeneity of profession was maintained to facilitate rapport-building [26]. Platform-assigned pseudonyms maintained participant anonymity.

The AOFG was conducted over 15 days, with 1 or 2 questions posted each day. An open-topic thread was created for discussing outstanding topics or providing feedback [29]. Participants could view and post responses at any time. Although responses were optional, participants were encouraged to reply to all questions and engage others in discussion. Participants’ responses were downloaded from the platform immediately after the AOFG’s conclusion.

Moderator interactions were clearly defined to avoid potential introduction of bias. They included twice-daily checks for participation or technical difficulties, posting follow-up questions, and removing any rule-breaking responses. Notifications and reminders for AOFG participation were disseminated on Days 8, 12, and 15.

### 2.6. Data Analysis

Thematic analysis was conducted using Braun and Clarke’s 6-phase framework [53,54], taking semantic and latent-level approaches [53,55], and open coding to facilitate the inductive identification of themes [55]. Transcripts were read repeatedly for familiarization. Themes were iteratively identified and coded by 1 independent coder with consensus among all authors. They were then reviewed by all 3 authors for the refinement and identification of subthemes and relationships, and then named and defined by discussion and consensus. An Excel spreadsheet was used for coding and thematic analysis.

## 3. Results

Participants comprised five public hospital pharmacists and two community pharmacists. The completion rate for the 12 main guiding questions was 89.3%. One participant answered only Questions 1–4. Hence, agreement from six or more participants constituted group consensus.

Two minor themes describing participants’ knowledge and management of MD and three major themes reflecting participants’ attitudes were derived (Figure 2). Corresponding exemplar quotes are shown in Table 2, Table 3, Table 4, Table 5 and Table 6.

### 3.1. Minor Theme: Knowledge about MD

Participants estimated that the prevalence of MD was low (<5%), based on the patients seen in practice, but believed it to be higher after taking children into consideration. There was an assumption that children and the elderly experienced MD to a greater extent than adults. Most participants listed altered pharmacokinetic profiles, increased risk of adverse effects, and lower efficacy as potential implications of DFM; few discussed incomplete dosing.

In addition, participants believed that patients crush medications as they are unaware of the potential implications. Some believed MD patients would otherwise be non-adherent, and listed patients’ perceived benefits of medications as determinants of adherence.

### 3.2. Minor Theme: Management of MD

#### 3.2.1. Sub-Theme: Management Strategies

Most participants expressed confidence in medication-related management of MD. Participants’ main strategies were split between DFM and switching to alternative formulations, which were usually liquids. Some mentioned other management strategies, such as switching to alternative drugs, speech language therapists (SLTs) referrals, reassurances, and patient education, but expressed unfamiliarity with training patients to swallow SODFs.

Participants partially agreed with the MD management plan (Figure 1). Most opined that the evaluation of swallowing difficulties fell outside their scope of practice as pharmacists and were not in favor of compounding. They also proposed the re-sequencing of steps on the MD management plan (Figure 1), but the suggestions were varied and sometimes contradictory.

#### 3.2.2. Sub-Theme: Tools in Management

When seeking information on MD management, most participants preferred consulting references over colleagues. Product inserts and in-house guidelines were frequently mentioned by participants to determine DFM suitability. Product inserts provide information and recommendations from manufacturers. They are objective in that when modification (e.g., crushing or chewing) is not recommended, the information is specified. Though most participants felt confident providing medication-related information, they pointed out that doctors or SLTs were needed to provide expertise on the assessment of patients; therefore, no single profession suited the role of information provider.

#### 3.2.3. Sub-Theme: Challenges in Management

Participants cited lack of expertise in evaluating swallowing difficulties, as well as time constraints for screening and managing MD as challenges encountered when managing patients with MD. In addition, they were further encumbered when pertinent information such as DFM suitability was missing from references.

When attempting to switch formulations, unavailability was the most frequently raised challenge. Few participants discussed the unpalatability, cost, and potentially different pharmacokinetic profiles of alternative formulations. One participant recounted experiencing prescriber resistance to switching formulations.

### 3.3. Major Theme: Expectations of Patient Proactivity

#### 3.3.1. Sub-Theme: Awareness of Greater Capacity

There was group consensus that MD was most likely identified during medication administration. More participants agreed that earlier identification of MD could occur during prescribing (*n* = 6) than dispensing (*n* = 4). Some believed that current practices were “reactive”, acknowledging more could be done in identifying MD patients.

#### 3.3.2. Sub-Theme: Reactive Approach of HCPs

Most participants did not screen for MD, but relied on patients and caregivers to notify them. Some participants performed conditional screening for CD—at transitions of care, when dispensing medications for children, or formulations that must be swallowed whole. Participants expressed belief that patients experiencing MD would proactively seek out HCPs, self-educate, or consult friends and family.

### 3.4. Major Theme: Desire for Objectivity

#### 3.4.1. Sub-Theme: Greater Emphasis on Objective Information

Participants preferred to seek objective information when identifying and managing MD patients. Responses to the scenario-based questions centered on seeking objective information, such as the medications involved, potential spoilage, or factual recounts of events, even when the scenario involved subjective factors (aversion). Objective factors of MD, such as product size and medical history, were discussed more frequently than subjective factors, such as psychological issues and misconceptions.

In the scenario-based question, when informed of the patient’s history of stroke, most participants altered their approach by adding referrals to SLTs.

#### 3.4.2. Sub-Theme: Lower Perceived Importance of Subjective Issues

Most responses reflected a lower perceived importance of MD by participants or their institutions, relative to CD or other tasks. Though infrequent, most participants expressed skepticism when discussing subjective factors or MD. Notably, one participant consistently referred to MD as “subjective difficulty”, despite referring to CD as “clinical dysphagia”.

### 3.5. Major Theme: Professional Roles

#### 3.5.1. Sub-Theme: Distinct but Complementary Roles

Participants limited their responses about MD management and information provision to medication-related issues. Most adopted a team-centric approach in managing MD by leveraging other professions’ expertise, sharing responsibility, or utilizing team-based decision making. This was clearly expressed by the switch from first person singular pronouns to plural pronouns “we” when discussing HCPs’ shared responsibility for patients.

Some participants would direct patients to ask other HCPs, such as doctors, whom they felt could manage MD better. The same participants highlighted interprofessional communication difficulties, which one participant attributed to their practice setting (retail pharmacy).

Few were informed of patients experiencing MD by colleagues. Most would only communicate this information to colleagues within their profession; one participant would not at all, though this was attributed to being downstream in the information flow.

#### 3.5.2. Sub-Theme: Concerns about Liability

Participants implied that checking for DFM suitability was their responsibility, as evidenced by the frequent mention of consulting references to determine DFM suitability. However, they believed that warning patients against DFM was sufficient to safeguard themselves from liability in case of patient harm resulting from DFM.

Some participants suggested patient education, obtaining prescriber’s approval for DFM, using available alternatives, and checking patient understanding as part of due diligence. One participant expressed willingness to raise minimum standards to include patient education if allotted more time. Another disagreed and reasoned that screening for MD exceeded the minimum standards required.

Most participants agreed that DFM done of the patient’s volition would absolve HCPs from liability in case of patient harm. Some mentioned DFM done against HCPs’ verbal or written advice as a disqualifier.

## 4. Discussion

Responses collected from the AOFG provided insights into pharmacists’ knowledge about MD, current practices in caring for MD patients, and attitudes underlying their practice. Attitudes towards professional roles established the boundaries within which participants managed MD. Expectations of patient proactivity, desire for objectivity, and assumptions about MD’s prevalence account for the limited screening for MD and underutilization of certain management strategies such as patient education and reassurance. Participants seemingly preferred DFM despite liability concerns, addressed by checking references for DFM suitability.

### 4.1. Knowledge

Knowledge about MD may be improved. Participants’ estimated prevalence (<5%) is much lower than the 10–40% suggested by the existing literature [4,5], and were influenced by their assumptions about affected demographics. Limited screening could have contributed to participants’ low perceived prevalence of MD, as postulated in other studies [56]. A study found that nearly half of its participants thought that patients aged 6–11 years were capable of swallowing SODFs with little to no difficulty [57], whereas other studies found decreasing trends of MD with age [1,4]. Additionally, the presence of MD in adults has been reported in various studies [1,12,13]. Future studies beyond the community setting are needed to improve the understanding of MD’s prevalence, and its presence across all age groups should be recognized and addressed.

Participants recognized DFM and non-adherence as common coping strategies adopted by patients with MD [1,4]. They were knowledgeable about various implications associated with DFM and did not limit discussion to modified-release preparations. This suggests participants avoided the pitfall of assuming DFM suitability in all immediate-release SODFs, unlike HCPs in other studies [4,20,56]. However, incomplete dosing appeared overlooked in discussions and is a potential pitfall, especially for medications with narrow therapeutic windows [4,16].

### 4.2. Management of MD

The management of MD by participants was bound within the clearly defined scope of practice as pharmacists. Suggested changes to the MD management plan were inconsistent or contradictory, highlighting the need for a standardized MD management plan. Participants consistently suggested delegating the evaluation of swallowing difficulties to doctors or SLTs, emphasizing their respective distinct but complementary roles. Compounding was not preferred, possibly due to greater resources required. As a result, it is seldom performed in local retail pharmacies, unlike overseas community pharmacies where it is more commonly offered [5]. In addition, medication review and removal of unnecessary medications not suitable for DFM were overlooked by participants as MD management strategies, despite them being within pharmacists’ scope of practice [4].

Participants’ suggestion that the identification of MD occurring earlier, during the prescribing phase, reflects assumptions that prescribers would have considered MD, so no further review by pharmacists is required.

DFM is participants’ preferred strategy in managing MD, possibly due to the challenges associated with switching formulations, the other strategy raised, including cost and unavailability [4,5,19]. However, being the last-resort strategy, DFM is not ideal [18]. Nonetheless, though switching formulations is recommended, HCPs should be mindful to check for dose equivalence or make necessary dose adjustments [5,18] for patient safety.

Other less commonly discussed strategies are worth further exploration. Given that psychological barriers are a commonly cited cause of MD [1,5,14,58], provision of reassurance seems underutilized. Participants’ desire for objectivity may have led to subjective factors being overlooked and unaddressed. However, severe cases dubbed “psychogenic dysphagia” require psychological intervention [5,19,59]. Training patients to swallow SODFs has also been suggested in other studies [1,4]. Despite evidence of efficacy in children and adolescents only [1,4,5], participants did not limit this strategy to children, as assumed in other studies [14]. Further studies proving efficacy in adults are required. Formulation-specific postural adjustments were discussed. Chin-tuck for medications that float on water [23] was effective in easing SODF administration in healthy persons and MD patients, but not necessarily in CD patients [1,5,60] due to aspiration risk; recommendations should be withheld until after evaluation [5].

In contrast, certain strategies could be avoided. Advising patients to take SODFs with sufficient water is probably ineffective despite being a commonly adopted coping strategy by MD patients [13,23], as insufficient water intake has been ruled out as a cause of MD [1]. Backwards-head-tilt for tablets is not recommended despite being intuitive and commonly practiced by MD patients [1,13,14,23]. It paradoxically increases swallowing difficulty [13,23,24] and aspiration risk [5]. Pharmacists should avoid recommending it, and actively warn against it.

Lack of time was a barrier to screening, management, and participants’ desire to do more for patients with MD. Manpower shortages may explain participants’ lack of time [61,62]. Alternatively, participants prioritized other tasks above MD management, having perceived it as less important given its subjective nature. Participants’ desire for objectivity was also reflected in their preference for consulting references over colleagues, regardless of practice settings, when in doubt about DFM suitability.

### 4.3. Expectations of Patient Proactivity

Expectations of patient proactivity to notify pharmacists of MD were reflected by limited screening for MD despite its capacity for earlier identification and awareness of patients’ potential coping strategies. Participants’ limited screening for MD, similar to findings from existing studies [1,13,56], may also be influenced by their assumptions about affected demographics. Conditional screening seems limited in scope and unlikely to spot previously unidentified MD patients. Pharmacists should be made aware that MD occurs across different age groups [1,12] and their breadth of screening should be improved.

On the other hand, studies have reported that, out of embarrassment or belief that HCPs would not be able to help [1,4], most patients do not notify HCPs if they experience MD; instead, advice from potentially less well-informed friends and family [4] would be sought. Patient education may correct misconceptions and improve patients’ perception of pharmacists and other HCPs as an approachable and trusted source of help and health information. Additionally, studies recommended HCPs proactively screen for MD [1,18,23]. The recently developed PILL-5 provides a solution for MD screening [9]. Comprising only five questions on patients’ experience with taking SODFs, as well as how they take them, it is simple to use, validated, and provides score-based recommendations, including when referral to specialists is necessary [9]. Being patient-administered, it circumvents time constraints and removes the need for evaluation skills, allowing HCPs of various professions to screen for MD rapidly. Alternatively, simply asking if patients have difficulty swallowing medications was also recommended [4,13,23].

### 4.4. Desire for Objectivity

CD was frequently discussed alongside objective factors, such as predisposing medical history, which may account for participants’ greater perceived importance of CD than MD. By assigning greater weight to objective factors, other common causes of MD such as anxiety and aversion to taking medications [1,5] could be overlooked, even if mentioned in patient interactions, leading to missed opportunities in identifying MD patients.

Expression of skepticism when discussing subjective factors and MD suggests a lower perceived importance of both. When answering the scenario-based question, participants’ revision of management strategies after objective factors were introduced suggests they started to take MD more seriously, substantiating the aforementioned theory. Participants’ perceived importance of MD may also be influenced by their respective places of practice [4]—infrequent communication about MD may lead to impressions that MD is a non-issue. Notably, few participants had learned of patients experiencing MD from colleagues, suggesting their colleagues did not regard MD important enough to screen for, or a lack of formal communication channels.

Ultimately, MD affects the medication use experience, which is invariably subjective, but may affect adherence and health outcomes [63].

### 4.5. Professional Roles

Despite participants’ views that MD management fell within the scope of practice of multiple professions, their clearly defined boundaries did not pose a barrier to MD management. Instead, participants recognized the complementary contributions required of each profession.

However, though the team-centric approach was desired, it was not applied in practice with regard to communication about MD patients, as most participants limited their communication to colleagues within the profession. Notably, some participants mentioned interprofessional communication difficulties, which may explain why they would direct patients to ask other HCPs instead of consulting on patients’ behalf. However, this practice places expectations of proactivity on patients. Encouraging interprofessional communication may improve the continuity of care [4,11]. For instance, notifying prescribers of patients’ MD may prompt a reconsideration of potentially inappropriate prescriptions and an evaluation of swallowing difficulties at subsequent appointments. More effort could be directed towards community settings, where lack of proximity was a postulated barrier to multidisciplinary discussion [56].

Prescribers’ resistance to switching formulations, although uncommon, suggests “power imbalance” in decision-making and that certain HCPs are not truly comfortable with a team-centric approach [64].

Participants opined that the provision of administration instructions, whether verbal or written, constituted minimum standards of due diligence, and believed that DFM of patients’ own volition would absolve HCPs from liability in case of harm. However, they did not consider the possibility of patients misunderstanding or overlooking written instructions. Emphasizing verbal warnings against DFM and checking patients’ understanding could minimize inappropriate DFM by patients and potential harm, but requires time and greater HCP proactivity.

Participants rarely discussed obtaining prescriber approval prior to recommending DFM, highlighting it as a potential pitfall as reported by previous studies [19,20]. Notably, they frequently discussed checking for DFM suitability alongside MD. They desired objectivity to provide assurance about liability concerns regarding DFM suitability, preferring products inserts as information sources, since they provide objective information about DFM suitability directly from the manufacturer.

### 4.6. Implications on Clinical Practice

This pilot AOFG provided insight into pharmacists’ KAP in caring for patients with MD. There are a few implications drawn from the findings. Firstly, SODF modification was among pharmacists’ preferred MD management strategies, and objectivity was desired. This points to the necessity to make references and information resources on the suitability of SODF modification available to their everyday practice. For pharmacists working in resource-constrained institutions, the INGEST algorithm [65] can be used as an alternative tool to guide them in making such evaluations. Being an implicit tool, it prompts users to consider attributes of SODFs and patients’ need for tube-feeding before determining their suitability for DFM [65]. Secondly, there appeared to be a lack of formal communication channel for conveying information on patients’ swallowing ability and/or needs for DFM. Establishing a standard communication channel will facilitate the timely transfer of such information within and between healthcare institutions.

### 4.7. Metholodogical Considerations

The use of the AOFG format in this study presented several benefits. It allowed participants to share their lack of knowledge or confidence, which might not have happened in face-to-face formats. Flexibility of time likely contributed to high completion rates, despite responses being optional. On the other hand, participants’ contributions were variable—those with fewer logins tended to post shorter, matter-of-fact responses. Additionally, most responses were posted on weekends. Future work should include more open-ended questions and ensure buffer-time falls on a weekend.

Some limitations of the AOFG format were identified in this study. Firstly, due to the asynchronous nature of the study, immediate feedback from other participants was not required. This made it challenging to ensure that the discussion was on track and that each participant considered others’ views when providing their opinions. In future AOFGs, more regular and targeted reminders, as well as a tighter deadline to respond (e.g., of 3 days instead of 2 weeks) could be introduced. Secondly, as there was no face-to-face or verbal interaction, the AOFG format lacked nonverbal cues and tone of voice, which can be very important in interpreting a message. This was minimized in part by regular follow-up questions posted by the moderator for clarification. Transcripts were also read repeatedly for familiarization and to capture the important points raised.

The interview guide was purposefully kept generic to avoid excluding or favoring any particular profession. In addition, to limit reflexivity and the introduction of personal biases from research team members, the discussion guide was structured with clearly defined moderator interactions. Independent coding, the identification of themes, and discussion to obtain consensus ensured reliability of findings [66]. The saturation of ideas was not achieved in this pilot study, but would likely be achieved with one AOFG for each healthcare profession [67]. Future plans include the recruitment of seven to nine members from each of the other professions (doctors, nurses, and SLTs) and the continued use of the AOFG format to elicit information on their KAP on caring for patients with MD. The insights gathered will help expand the evidence base on the management of MD and contribute to the ongoing research on improving care for patients with this condition.

## 5. Conclusions

This pilot AOFG provided insight into pharmacists’ KAP in caring for patients with MD. Knowledge about MD could be improved, especially regarding prevalence-related assumptions. Preferred MD management strategies were DFM and switching formulations, each with associated challenges. Expectations of patient proactivity were reflected by limited screening for MD, despite recognizing opportunities for the earlier identification of MD. Pharmacists’ desire for objectivity possibly accounted for their apparent lower perceived importance, resulting in overlooking the subjective factors of MD and corresponding management strategies.

Clear boundaries between professional roles did not appear to pose barriers to MD management, which was deemed a multidisciplinary task. Providing administration instructions warning against DFM was defined as the minimum standard to safeguard against potential liability in case of patient harm. However, the findings are limited since the AOFG was conducted with pharmacists and may not be generalizable beyond this group.

The insight into pharmacists’ KAP and methodological considerations may be incorporated into the full-scale study involving participants across various healthcare professions. Future work may also include studies on MD’s prevalence across various care settings and the development of MD management plans.

## Figures and Tables

**Figure 1 ijerph-20-02858-f001:**
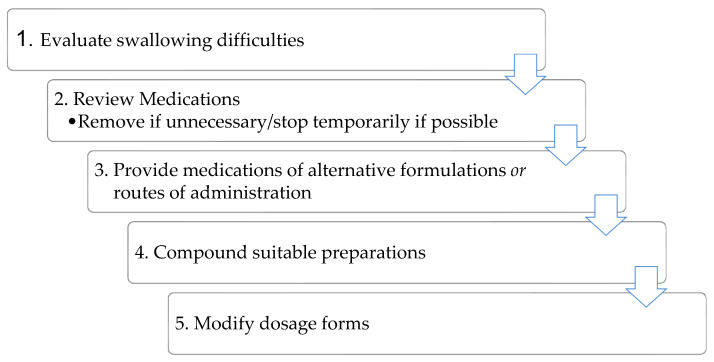
Medication dysphagia management plan summarizing common recommendations.

**Figure 2 ijerph-20-02858-f002:**
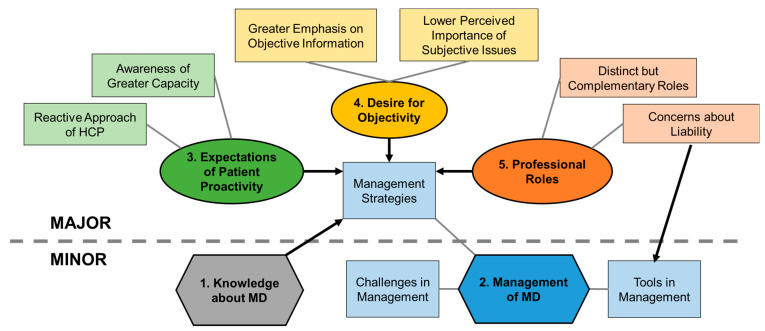
Thematic map of five themes derived from the thematic analysis. Major themes reflecting attitudes in MD management are represented by ovals. Minor themes describing knowledge and practices in MD management are represented by hexagons. Subthemes are represented by rectangles and connected to main themes by plain lines. Relationships between themes are represented by arrows.

**Table 1 ijerph-20-02858-t001:** Features of the AOFG format and corresponding postulated benefits.

Features of the AOFG Format	Postulated Benefits
Participation from any location with internet access	Minimization of health and safety risks amidst a pandemic
Participants can respond at a time of their choice/convenience during the study	Flexibility of time for potentially better response rates [26,28,29]
	Elicitation of more thoughtful responses by removing the need for an immediate answer [26,29]
Participant anonymity	Minimization of social desirability bias and power imbalances to disinhibit participants and facilitate honest discussion [26,28]
Direct input of data by participants	Improved accuracy and time-savings in data collection [26,28,29]
Direct download of participants’ responses	

**Table 2 ijerph-20-02858-t002:** Minor theme of knowledge about medication dysphagia and corresponding exemplar quotes.

Theme	Topic	Exemplar Quotes (Participant)
Knowledge about medication dysphagia	Low prevalence	“Maybe about 5% … I don’t think this is representative of Singapore’s population as I rarely see children in my practice” (#7)
	MD assumed in children and elderly	“Most common group: children…medication dysphagia is relevant in virtually all pediatric patients I see…2nd most common group: elderly patients…” (#3)
	MD assumed absent in adults	“In adults…we generally assume patients has no difficulty swallowing medications until they voice out the problem.” (#4)
	Temporary nature of MD in children	“Definitely most common in children. Sometimes this carry through to teens and adulthood” (#5)
	Persistent nature of MD	“If … there’s no durable alternative for patient, we can…” (#4)
	Coping strategies of DFM and non-adherence Patients unaware of implications of DFM	“I agree that patients may just crush the medication, cut it, attempt to dissolve the tablet or open up the capsules. I have also encountered some anecdotal evidence from caregivers that some patients just do not take the medication. Patients are likely not cognizant that the actions of modifying the medication may alter the intended PK properties of the medication, increase adverse effects or even render the medication less or ineffective.” (#5)
	Patients’ perceived benefits and non-adherence	“Depends on how the patient view the medication. (1) Patient does not view the medication as something ‘real’ or ‘important’…more likely that patient will just default their medication” (#1)

**Table 3 ijerph-20-02858-t003:** Minor theme of management of medication dysphagia, corresponding subthemes, and exemplar quotes.

Theme	Subtheme	Topic	Exemplar Quotes (Participant)
Management of Medication Dysphagia	Management Strategies	Confidence in managing MD	“I am confident of caring for patients with medication dysphagia…” (#5)
		DFM and switching formulations	“… figure out if there’s anything that can be done before considering alternative agents e.g., tablet too large -> cut or crush the tablets if possible, or find a liquid product if possible.” (#3)
		Other MD management strategies	“The last resort might be to switch to another drug within the same class…” (#7) “…team would assess and decide if patients needs referral to speech therapist…” (#4) “…I would take a reassuring and encouraging stance” (#6) “…tilting the head slightly back helps with tablets and tilting slightly forward helps with capsules. Not sure how useful this is since I never really had the chance to try advising this in actual practice or to search for corroborating references.” (#3) “Is there a way to “train” individuals to swallow pills?” (#4)
		Partial agreement with MD management plan	“While I agree with the algorithm in principle, I find it hard for me as a pharmacist to properly evaluate a patient’s swallowing difficulties…but don’t consider compounding to be a realistic option” (#3)
	Tools in Management	Consulting references for DFM suitability	“(i) Product inserts is a good source of information that pharmaceutical companies can guide how the medication can be handled (ii) institutional/in house guidelines would have a collated list of medications and alternatives to recommend i.e., how to modify to accommodate patients with med dysphagia” (#6)
		Consulting colleagues	“I would turn to seniors when in doubt… to pharmacists who are handling HIV clinics, or oncology patients or even contact pharmacists at our cancer center pharmacy” (#4)
		Expertise of other professions required	“I would feel that as pharmacists, we would be suited to advise on the medication administration and formulation concerns. But speech therapists would be in a better position to evaluate and manage the patient.” (#6)
	Challenges in Management	Lack expertise in evaluation	“I agree with the other participants that as pharmacists, we are not trained to evaluate swallowing difficulties” (#5)
		Lack of time	“I don’t usually screen for medication dysphagia …also because it takes up too much time, especially when I have to check for other medical/medication history. It takes more time to manage their medication dysphagia problems as well” (#3)
		Missing information from references	“…in-house NGT guide for common medications whether they can be crushed and possible alternatives. However, the list is non-exhaustive and not updated frequently.” (#4)
		Availability issues	“…liquid alternatives are usually available in varying rarities…” (#3)
		Different pharmaco-kinetic profile of alternative formulations	“provide medications of alternative formulations (if available) and/or routes of administration (even though the bioavailability and onset of action of medication may be altered).” (#6)
		Prescriber resistance	“I once had trouble convincing a prescriber to consider alternatives for a patient…” (#3)

**Table 4 ijerph-20-02858-t004:** Major theme of expectations of patient proactivity, corresponding subthemes, and exemplar quotes.

Theme	Subtheme	Topic	Exemplar Quotes (Participant)
Expectations of Patient Proactivity	Awareness of Greater Capacity	Earlier identification of MD	“I think it is most likely to be discovered in the administering stage.” (#2) “Yes, if HCP could identify during the prescribing or dispensing phase by asking the patient or caregiver if there’s any difficulty or issues with medication…” (#4)
		Acknowledging more could be done	“Currently, we are still more ‘reactive’ in identifying them.” (#1) “I have never thought of how to guide patient in identifying medication dysphagia. Hopefully there are validated questions out there that is easy to administer. (I have yet to do my lit-search).” (#1)
	Reactive Approach of HCPs	Reliance on patient/caregiver reporting	“Unfortunately, I don’t actively screen for medication dysphagia in these patients. It may not be something that actively comes to my mind in a busy day. I will address it if patients or family members bring up the issue.” (#2)
		Conditional screening for MD	“The only time where we will actively ask is when I am dispensing the medication for a child below 12 years old.” (#1) “I find out when I ask them regarding any medication dysphagia, if I am dispensing medications that must be swallowed whole. For med recon inpatient, the presence of medication dysphagia is routinely asked as well.” (#7)
		Extended expectations of patient proactivity	“Patients probably aren’t aware of such implications as they weren’t informed, though some may discuss with a HCP or google about their medications then they could be aware” (#4) “They would … or simply check with a family member.” (#6)

**Table 5 ijerph-20-02858-t005:** Major theme of desire for objectivity, corresponding subthemes, and exemplar quotes.

Theme	Subtheme	Topic	Exemplar Quotes (Participant)
Desire for Objectivity	Greater Emphasis on Objective Information	Seeking objective information	“Find out more details regarding his/her past experience with medication dysphagia e.g., -what medication/supplement caused the problem -what was the problem exactly e.g., burning sensation in the throat after swallowing doxycycline, problems swallowing calcium supplements” (#3)
		Subjective factors of MD	“Well, it can happen earlier if the tablet is large…” (#6) “…if I can see medical records and see medical conditions that may predispose a patient to medication dysphagia, this will help me identify patients…” (#3)
			“Came across patients with psychological issues that exhibit this.” (#5) “Subsequently, there may be some effort in determining if there is a concern about ‘taking medication’ (negative association with medicine, such as medication damaging the kidney, being addicted, being reliant, etc).” (#1)
		Change in approach after obtaining additional objective information	“One thing I would additionally advise the patient is to ask the doctor managing his/her post-stroke care regarding possibility of seeking the expertise of a speech therapist for managing the dysphagia” (#3)
	Lower Perceived Importance of Subjective issues	Prioritization of other tasks and issues above MD	“I don’t usually screen for medication dysphagia…because it takes up too much time, especially when I have to check for other medical/medication history.” (#3)
			“Most of the emphasis is placed on clinical dysphagia. Come to think of it, there are not much communication about medication dysphagia.” (#1)
		Expression of scepticism to subjective factors and MD	“…there is a ‘psychological’ barrier…” (#5) “Given the recent stroke, the possibility of having a ‘real’ medication dysphagia/clinical dysphagia is higher” (#1)

**Table 6 ijerph-20-02858-t006:** Major theme of professional roles, corresponding subthemes, and exemplar quotes.

Theme	Subtheme	Topic	Exemplar Quotes (Participant)
Professional Roles	Distinct but Complementary Roles	Kept within scope of practice	“caring for patients with medication dysphagia—through seeking alternatives in liquid preparations and crushing of solid preparations” (#5)
		Team-centric approach	“…I would recommend that each step should be performed by different HCP, such as step 1 can be performed by Dr or speech therapist as they would be more appropriate to evaluate the patient’s swallowing difficulty…Modify dosage forms: Performed by nurses where they are the ones administering the medications…” (#4)
			“I will be comfortable providing information to colleagues about the medication-related needs… I think doctors are best placed to assess this problem, though as mentioned, a multi-disciplinary involvement can be the final model.” (#5)
		Direct patients to ask other HCPs	“…I would just recommend discussing this with the dr” (#3)
		Inter-professional communication difficulties	“more limited access to communicate with prescribers” (#3)
		Limited inter-professional communication	“I usually find out…through the SOAP notes/counselling notes of doctors/pharmacy colleagues” (#5) “Similarly I will include the issue and management under counselling remarks to keep my pharmacy colleagues in the loop” (#5) “Usually I’m among the last to find out, so there’s no need to communicate this information.” (#3)
	Concerns about Liability	Administration instructions as minimum standards	“Due diligence probably ends at the point where the information of ‘swallow whole, do not crush’ is being presented to patient.” (#1)
		Checking DFM suitability	“Crushing tablets will only be suggested if it is stated in the PIL. Most people are concern of recommending any action that is not stated in the PIL.” (#1)
		Obtaining prescriber approval	“…we should do our due diligence to run a check with the physicians to ensure a concurrence that modification of medications are made to accommodate the patient…” (#6)
		Raising minimum standards	“However, if sufficient consideration is given to the capacity of the pharmacist to provide more detailed counselling (allow more time per consultation session), then I may be receptive to higher standards of due diligence…” (#3)
		Screening for MD beyond minimum standards	“Going the extra mile perhaps meant asking the patient actively: do you think you will have problem swallowing the tablet.” (#1)
		DFM of patient’s volition	“However, if the patient decide to modify the medication without discussing with the healthcare professional, or modifying the medication even it is advised against, then it is at their own risk.” (#1)

## Data Availability

The data presented in this study are available on request from the corresponding author. The data are not publicly available due to their containing information that could compromise the privacy of research participants.

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
