# Peer review of "A Pilot Study on Pharmacists’ Knowledge, Attitudes and Practices towards Medication Dysphagia via Asynchronous Online Focus Group Discussion"

_ijerph, 2023, doi:10.3390/ijerph20042858_

Round 1

Reviewer 1 Report

Thank you for the opportunity to review this manuscript. This study aims to investigated pharmacists’ knowledge, attitudes, and practices (KAP) in caring for patients with medication dysphagia. There is scant information on this topic in the literature, yet the underlying prevalence of this problem in real world settings might be more pronounced than what we currently know.

This is a meaningful study, though the interpretation of the results is limited by the small size and a (hardly) representative group of participants. The findings synthesized from the focus group is probably reflective of pharmacists' perception... but not other HCPs. Hence, I suggest that authors should keep this point in mind when making inferences in the Discussion section. I also have some questions on the methodology.   

Introduction:

1. I think the Introduction provides adequate information on MD. However, it can be further improved if the authors can consider adding more information on the role of pharmacists in managing AD as this focus group study only involved pharmacists. 

2. Line 82 "Studies on MD have focused on patients’ perspectives and shed little light on HCPs’ knowledge, attitudes, and practices (KAP)" I also think that Introduction may be better structured according to the concept of KAP, since this is the focus of the study. 

3. Line 83 "This study aimed to conduct a pilot study on pharmacists to investigate their KAP in caring for patients with MD." Since this is a pilot study, this should be reflected in the title of the study. 

Methods:

1. Line 110: 'Inclusion criteria included 110 current involvement in direct patient care"- are pharmacists from all sectors recruited for the study? I think the issue of MD might be more relevant for a pharmacist working in a nursing home or hospitals or the community?

2. Line 116. I apologize if I have misinterpreted the authors. The authors stated "convenience sampling". So I thought that the invitations should be sent to all potential eligible pharmacists (perhaps via mass emailing or advertisment). However, I read on and Line 116 says " Of the 9 pharmacists invited..." this seems more like purposive sampling. May I ask if the authors can clarify on the sampling approach? 

3. Are there more than 1 independent coders? Any qualitative analysis software used for this study?

Results:

The results are generally clear and well written. May I ask if the authors can clarify on the following?

1. Line 157: "participants believed that patients modify medications as they are unaware of potential implications" - how do the patients "modify medications"?

2. Line 196. Suggest to change to "....prescribing (n = 6) than dispensing (n = 4)

Discussion

1. Overall, the Discussion section included relevant discussion. However, the organization can be improved. There are some abrupt paragraphs that consist of just 1 to 2 sentences (eg. line 294-295, line 330-332). Hence it is rather difficult to read. Suggest to deliver 1 to 2 key messages per paragraph for each section.

2. As mentioned earlier in this review report, this study only involves 7 participants who are pharmacists. And the inferences were based solely on these 7 individuals (and literature review). It seems rather unfair or unconvincing to say that "Participants’ suggestion that identification of MD occurring earlier, during the prescribing phase, reflects assumptions that prescribers would have considered MD, so no further review by pharmacists is required. This suggests that identifying MD patients is an “orphan task” – each profession does not actively screen for MD, assuming that another will [56]."  The views of the doctors were not sought in this study.

Similarly... Line 389 "Participants agreed that provision of administration instructions, whether verbal or written, constituted minimum standards of due diligence, while DFM of patients’ own volition absolved HCPs from liability in case of harm. These cut-offs overlook the fact that some patients may misunderstand or overlook written instructions".. Again, these inferences are strong statements because patients' perceptions are not directly evaluated in this study. 

3. The authors mentioned about the advantages of AOFG. However, there are distinct limitations of this approach. They should be discussed too. 

4. Since this is a pilot study, what are the future plans for the full scale study? This can be included in the Discussion too. MD is an important yet under-addressed topic, it would be nice to hear from the authors the implications of their current research on clinical practice. 

5. Line 151 'Participants estimated that prevalence of MD was low (<5%), based on the patients seen in practice, but believed it to be higher, after taking children into consideration."... It seems like we cannot refute the participants' experiences when they considered the prevalence of MD to be low (even though the literature says otherwise). I wonder if the authors would consider targeting the full scale study on HCPs that serve patients who are known to be at a higher risk for MD (eg. elderly, children etc.). This is just a suggestion - I apologize in advance if this is not the main goal or direction of your research work. 

Thank you again for the opportunity to review this manuscript. 

Reviewer 2 Report

Thank you for researching such an important topic.
